# Identifying interpretable gene-biomarker associations with functionally informed kernel-based tests in 190,000 exomes

Remo Monti[1,2], Pia Rautenstrauch[2,3], Mahsa Ghanbari[2], Alva Rani James[1], Matthias Kirchler [1,4], Uwe Ohler [2,5,7], Stefan Konigorski [1,6,7] & Christoph Lippert [1,6,7] ✉

Here we present an exome-wide rare genetic variant association study for 30 blood biomarkers in 191,971 individuals in the UK Biobank. We compare gene-based association tests for separate functional variant categories to increase interpretability and identify 193 significant gene-biomarker associations. Genes associated with biomarkers were ~ 4.5-fold enriched for conferring Mendelian disorders. In addition to performing weighted gene-based variant collapsing tests, we design and apply variant-category-specific kernel-based tests that integrate quantitative functional variant effect predictions for missense variants, splicing and the binding of RNA-binding proteins. For these tests, we present a computationally efficient combination of the likelihood-ratio and score tests that found 36% more associations than the score test alone while also controlling the type-1 error. Kernel-based tests identified 13% more associations than their gene-based collapsing counterparts and had advantages in the presence of gain of function missense variants. We introduce local collapsing by amino acid position for missense variants and use it to interpret associations and identify potential novel gain of function variants in *PIEZO1*. Our results show the benefits of investigating different functional mechanisms when performing rare-variant association tests, and demonstrate pervasive rare-variant contribution to biomarker variability.

Large biobanks that combine in-depth phenotyping with exome sequencing for hundreds of thousands of individuals promise new insights into the genetic architecture of health and disease[1]. Whilst common-variant association studies have detected tens of thousands of loci associated with heritable traits, the underlying functional mechanisms remain largely unknown due to linkage disequilibrium and the fact that the majority of loci lie in non-coding regions of the genome[2]. Furthermore, effect sizes of common variants tend to be small, as variants with large detrimental effects are selected against, which limits their frequency[3,4].

Association studies using whole exome sequencing (WES) do not face these issues to the same extent, as they largely contain more interpretable loci where an enrichment for large effect sizes is expected[5]. However, the majority of genetic variants identified by WES are extremely rare, and the vast number of these variants poses challenges for rare-variant association studies (RVAS), given the burden of

[1]Digital Health - Machine Learning, Hasso Plattner Institute, University of Potsdam, Digital Engineering Faculty, 14482 Potsdam, Germany. [2]Max-Delbrück-Center for Molecular Medicine in the Helmholtz Association (MDC), Berlin Institute for Medical Systems Biology (BIMSB), 13125 Berlin, Germany. [3]Humboldt-Universität zu Berlin, Department of Computer Science, 10099 Berlin, Germany. [4]TU Kaiserslautern, Department of Computer Science, 67663 Kaiserslautern, Germany. [5]Humboldt-Universität zu Berlin, Department of Biology, 10099 Berlin, Germany. [6]Hasso Plattner Institute for Digital Health, Icahn School of Medicine at Mount Sinai, New York 10029-6574 NY, USA. [7]These authors contributed equally: Uwe Ohler, Stefan Konigorski, Christoph Lippert. ✉e-mail: christoph.lippert@hpi.de

multiple testing and low statistical power due to low allele frequencies[6,7]. For these reasons, variants in RVAS are typically grouped into sets that correspond to functional units, such as genes, before association testing[6–9]. Not only does this strategy aggregate signal and thereby increase statistical power, but it also lessens the burden of multiple testing. Burden tests, for example, collapse variants within genes into a single variable prior to association testing, i.e., perform gene-based variant collapsing[6,10]. Alternatively, kernel-based tests aggregate groups of variants into a so-called kernel-matrix that is tested using a score test[8] or the likelihood-ratio test (LRT)[9] without the need for collapsing. Among these, the LRT has higher statistical power when effect sizes are large, but is computationally more expensive[11,12].

While gene-based variant collapsing performs best in the presence of many causal variants with effect sizes that point in the same direction (e.g., increasing risk for disease), kernel-based tests have advantages in cases of opposing effects and fewer causal variants[7]. To increase the fraction of causal variants, exome-wide RVAS that use variant collapsing have defined qualifying variants based on annotations such as allele frequencies or variant effect predictions and excluded all other observed variants from the association tests[6,13–16]. These studies have mostly focused on non-synonymous variants, where software tools identify protein-truncating variants and

distinguish between benign and potentially deleterious missense variants[17–20].

Here, we perform an extensive RVAS using exome sequencing data from the UK Biobank[21]. For approximately 190,000 individuals, 30 quantitative biomarkers provide objectively quantifiable measures related to the health status of individuals[22], making them attractive phenotypes for genome-wide association studies[23]. We go beyond the collapsing tests for coding variants described above and explore the use of kernel-based association tests and deep learning-derived variant effect predictions for gene-regulatory variants, namely for splicing[24] and the binding of RNA-binding proteins (RBPs)[25].

Specifically, we use quantitative functional variant effect predictions to group and weigh variants in gene-based association tests and increase interpretability. The greater flexibility of kernel-based tests allowed us to design variant category-specific tests and combine collapsing and non-collapsing approaches in the same tests. For kernel-based tests, we show that a computationally efficient combination of the score test and the restricted likelihood-ratio test (RLRT) can identify 36% more significant associations compared to the score test alone. We find 193 significant gene-biomarker associations in total, the majority of which confirm associations previously reported to GWAS databases (87%)[26,27].

We find that including participants from diverse ancestries identifies 14% more associations than limiting to participants with strict inferred European genetic ancestry, while increasing the sample size by 17%. Associations with biomarkers frequently occurred in genes linked to Mendelian disorders, and comparisons to other association studies confirmed their plausible disease relevance. Finally, we interpret associations that were only found for specific functional variant categories or associations for which weighted gene-based variant collapsing and kernel-based tests gave vastly different results. This provided additional biological insights and highlighted the benefits of a diverse testing strategy for RVAS.

## Results
### Data description and workflow

We performed an RVAS of 30 quantitative serum biomarkers in UK Biobank 200k WES release[21]. These biomarkers contain established disease risk factors, diagnostic markers, and markers for phenotypes otherwise not well assessed in the UK Biobank cohort. They can roughly be grouped into cardiovascular, bone and joint, liver, renal, hormonal, and diabetes markers (Table 1). After removing related individuals and restricting the analysis to those with no missing covariates, 191,971 participants with diverse genetic ancestry remained. About 15,702,718 rare (MAF < 0.1%) variants were observed in this subset and passed quality criteria, including pruning variants with large deviations from the observed European MAF in other ancestries (Methods). The median sample size for the biomarkers was 181,784 and ranged from 15,997 (Rheumatoid factor) to 182,742 (Alkaline phosphatase). About 191,260 participants had at least one measured biomarker.

We used functional variant effect predictions to group and weigh variants and performed functionally informed gene-based association tests. Specifically, we chose to investigate strict protein loss of function variants (pLOF, e.g., frame shift or protein-truncating variants), missense variants, splice-altering variants, and variants predicted to change the binding of RNA-binding proteins (Fig. 1, Methods), the latter two originating from deep learning models[24,25].

We treated these categories separately during association testing to increase interpretability, resulting in multiple tests per gene, which we refer to as separate models. Specifically, we adapted either kernel-based tests, weighted gene-based variant collapsing, or both types of association test depending on the variant effect category (Methods). Tests were performed using a combination of score tests and restricted likelihood-ratio tests. We make variant effect predictions for all variants in the UK Biobank 200k exome release available (https://

**Table 1 | UK Biobank blood biomarkers analyzed in this study**

| category | biomarker | short | N | sign. genes |
|---|---|---|---|---|
| bone and joint | Alkaline phosphatase | ALP | 182,742 | 10 |
| bone and joint | Calcium | | 168,054 | 3 |
| bone and joint | Rheumatoid factor | | 15,997 | 0 |
| bone and joint | Vitamin D | | 174,473 | 4 |
| cardiovascular | Apolipoprotein A | ApoA | 167,050 | 18 |
| cardiovascular | Apolipoprotein B | ApoB | 181,808 | 6 |
| cardiovascular | C-reactive protein | CRP | 182,320 | 3 |
| cardiovascular | Cholesterol | | 182,735 | 9 |
| cardiovascular | HDL cholesterol | HDL | 168,043 | 20 |
| cardiovascular | LDL direct | LDL | 182,420 | 6 |
| cardiovascular | Lipoprotein A | | 146,249 | 13 |
| cardiovascular | Triglycerides | TG | 182,587 | 12 |
| diabetes | Glucose | | 167,916 | 3 |
| diabetes | Glycated hemoglobin | HbA1c | 182,494 | 9 |
| hormonal | IGF-1 | | 181,759 | 7 |
| hormonal | Oestradiol | | 30,343 | 0 |
| hormonal | SHBG | | 166,521 | 6 |
| hormonal | Testosterone | | 165,211 | 1 |
| liver | Alanine aminotransferase | ALT | 182,675 | 5 |
| liver | Albumin | | 168,139 | 4 |
| liver | Aspartate aminotransferase | AST | 182,096 | 2 |
| liver | Direct bilirubin | | 154,963 | 6 |
| liver | Gamma glutamyltransferase | GGT | 182,655 | 7 |
| liver | Total bilirubin | | 181,998 | 8 |
| renal | Creatinine | | 182,639 | 7 |
| renal | Cystatin C | | 182,720 | 7 |
| renal | Phosphate | | 167,808 | 3 |
| renal | Total protein | | 167,931 | 5 |
| renal | Urate | | 182,538 | 8 |
| renal | Urea | | 182,612 | 1 |

Biomarker categories, biomarker names, their abbreviations used in the text (short), sample size (i.e., participants non-missing phenotype and complete covariates, N) and number of distinct genes significantly associated with each biomarker (sign. genes) after multiple testing correction (FWER ≤ 0.05).

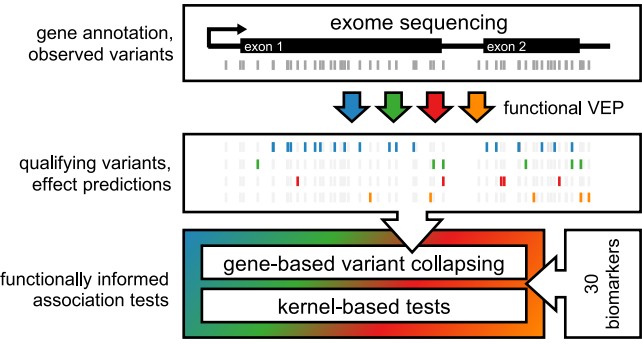

**Fig. 1 | Rare-variant association testing pipeline.** Exome sequencing measures exon-proximal genetic variants. All variants are subjected to functional variant effect prediction (VEP). Qualifying variants are determined based on the variant effect predictions and minor allele frequencies (MAF < 0.1%) and categorized based on their predicted functional impacts (protein loss of function, missense, splicing, RBP-binding). Finally, we test the different categories of qualifying variants in gene-based association tests against 30 biomarkers using gene-based variant collapsing and kernel-based tests.

github.com/HealthML/ukb-200k-wes-vep), as well as our analysis pipeline (https://github.com/HealthML/faatpipe) and software used to perform association tests (https://github.com/HealthML/seak).

## Functionally informed association tests

**Protein loss of function.** We predicted the effects of genetic variants on protein-coding genes using the Ensembl variant effect predictor[17] and found 463,479 pLOF variants with a median of 20 pLOF variants per gene (Fig. 2, Methods). For pLOF variants, we assumed a large fraction of potentially causal variants, and that variants within the same gene should, by and large, affect the phenotype in the same direction. For these reasons, we performed gene-based variant collapsing tests (Methods) and found 88 significant associations originating from 53 distinct genes.

**Missense.** We defined 1,834,082 high-impact missense variants based on PolyPhen-2[18] and SIFT[19] (Methods). 18,420 genes contained at least one high-impact missense variant, with a median of 73 high-impact missense variants observed per gene.

We hypothesized that missense variants in the same gene might have both trait-increasing and trait-decreasing effects, and that there might be fewer causal variants. Therefore, we did not only perform weighted variant collapsing tests for these variants, but also kernel-based association tests. We designed a missense-specific kernel that collapses variants locally by amino acid position, which affected 20% of variants. We further used the Cauchy Combination Test (CCT)[28] to dynamically incorporate pLOF variants in these tests (Methods). Combining missense and pLOF variants has been shown to increase the number of discoveries in RVAS, due to many missense variants leading to a loss of function[13,15,16]. We identified 146 significant associations using gene-based variant collapsing, and 160 using kernel-based association tests, with an overlap of 121. The total of 185 associations identified by either model originated from 101 distinct genes.

**Splicing.** We located 737,795 potentially splice-altering rare single nucleotide variants in 17,158 genes by cross-referencing against published SpliceAI variant effect predictions[24] (Methods). The median number of variants per gene was 30. We hypothesized that these splice variants could have complex downstream consequences and decided to compare both weighted gene-based variant collapsing and kernel-based association tests. For kernel-based tests, we used the weighted linear kernel[8]. As for missense variants, we dynamically incorporated pLOF variants in these tests to increase power.

We identified 75 significant associations with gene-based variant collapsing in 44 distinct genes, whereas kernel-based tests identified 88 significant associations in 51 genes. As our definition of pLOF variants included variants that directly hit annotated splice donor/acceptor sites, there was a considerable overlap of 95,842 variants between these annotations (21% of all pLOF variants). We therefore expected (and found) large overlaps (69, 74%) in the significant associations for pLOF and splice variants.

**RBP-binding.** Splicing is only one of several eukaryotic post-transcriptional regulatory mechanisms mediated by interactions of RNA-binding proteins (RBPs) with their target RNAs. As the UK Biobank WES data also contain variants in non-protein-coding parts of mRNAs, namely in introns (41.43%) and UTRs (11.32%), we reasoned that we may be able to identify variants with regulatory effects mediated by differential binding of RBPs. Specifically, we investigated if changes in the binding of RBPs predicted by DeepRiPe[25] could be associated with biomarker levels. The six RBPs QKI, MBNL1, TARDBP, ELAVL1, KHDRBS1, and HNRNPD were selected based on their binding preferences (introns, exons)[29], the high performance of the model to predict genuine target sites for these RBPs, and the reported presence of clear binding sequence motifs. We predicted variant effects for these RBPs and identified 370,880 variants with large predicted effects in 17,394 genes, with a median of 12 variants per gene (Methods).

As we expected a low number of causal variants and potentially opposing effect sizes, we only performed kernel-based association tests and identified nine significant associations in nine distinct genes.

## Integrative analysis overview

Merging the results from all models yielded a total of 193 associations originating from 117 distinct genes (Supplementary Data 1, 212 associations if counting genes with shared exons separately). We found at least one significant association for all but two biomarkers (oestradiol and rheumatoid factor) (Fig. 2 and Supplementary Figs. 1–3). For the majority of associations (174; 82%), tests combining missense and protein LOF variants gave the smallest p values.

We calculated the genomic inflation factor $\lambda_{GC}$ across all tests that were performed genome-wide, and did not find evidence of inflated type I error levels (Supplementary Fig. 8 and Supplementary Data 5). $\lambda_{GC}$ ranged from 0.84 to 1.06 for gene-based variant collapsing tests (median = 0.97) and from 0.92 to 1.08 for kernel-based tests (median = 1.01). Of the 117 distinct genes, 43 (37%) were associated with more than one biomarker, and a few genes had five or more significant associations: *ANGPTL3*, *APOB*, *JAK2*, *GIGYF1*, and *G6PC*. Many of the genes we found to be associated with biomarkers had either been implicated in diseases related to those biomarkers (e.g., *LRP2* with renal markers[30]) or are mechanistically related to the biomarkers themselves (e.g., cystatin C with its own gene, *CST*).

80% of genes (±5kb) contained single variants associated with the same biomarkers identified by a study using imputed genotypes and a larger sample size on the same data[23]. Yet, 97% of gene-phenotype associations remained significant after conditioning on the most significant variants identified by that study (considering variants ±500 kb around the gene start positions). We therefore conclude that most associations we observed are largely independent of the signals captured by array-based genotypes of (mostly) common variants.

## Including all ancestries increases the power for biomarker traits

We repeated our analysis restricting to 164,148 individuals of strict inferred European ancestry (Methods). While 175 associations were significant in both analyses, 37 were only identified in the analysis with all ancestries (AA) and 11 were only significant in the analysis with EUR individuals (Fig. 3 and Supplementary Data 3). The majority of hits significant in one analysis but not in the other were close to the significance threshold. Outliers such as the associations of *ALD16A1* with

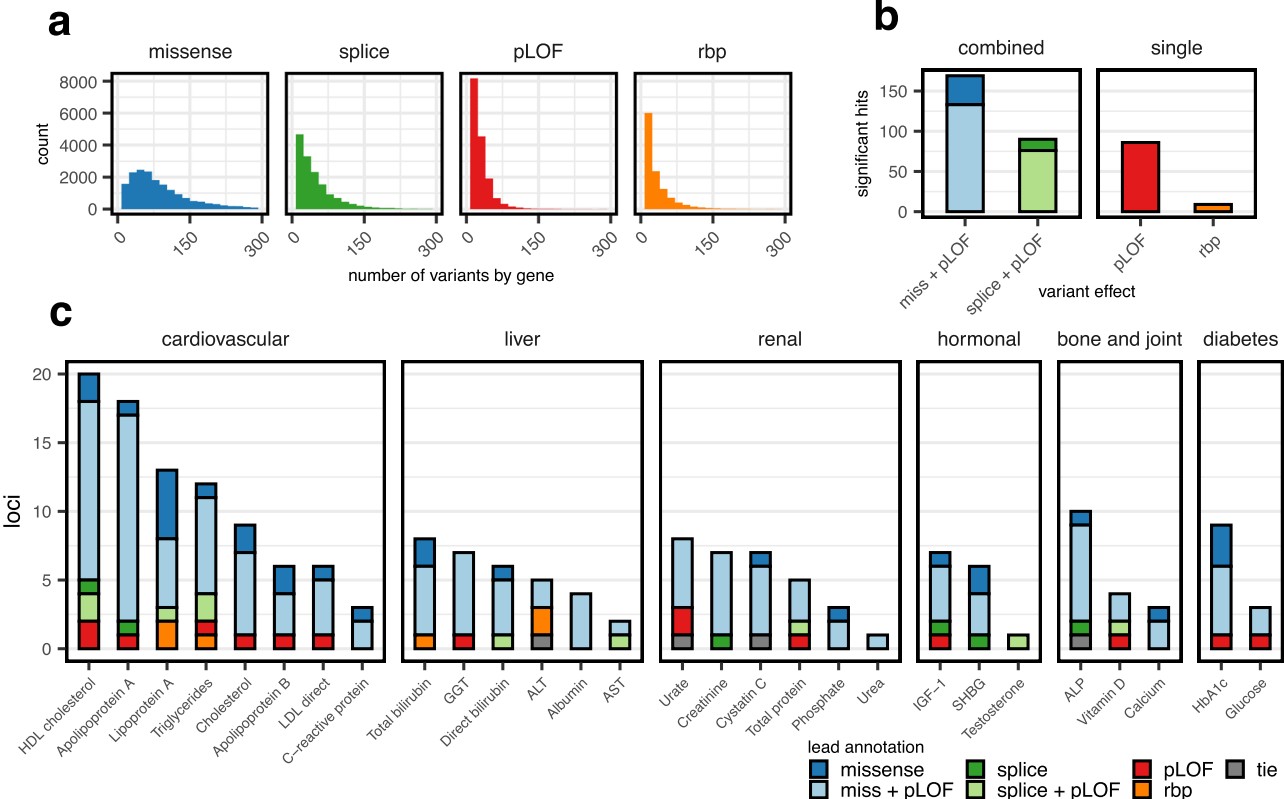

**Fig. 2 | Association tests overview. a** Histograms of the number of qualifying variants per tested gene for the different variant categories. Ranges are truncated at 300 variants, which affected 730 genes for missene, 84 for splice, 7 for pLOF, and 15 for rbp. **b** Bar plot of the number of significant genes found by testing qualifying variants in the different categories. Tests for splice and missense variants (left) were dynamically combined with pLOF variants, i.e., two *p* values, one arising from a test including only missense/splice variants and one combining those variants with pLOF variants were combined using the Cauchy combination test. **c** Bar plot showing 193 significant gene-biomarker associations for 28 biomarkers (x-axis). Bars in panels (**a**) and (**b**) are colored by the variant-type (or a combination thereof) which gave the lowest *p* value (lead annotation).

Urate could all be explained by hard thresholds for variant inclusion and strong signals from single variants. Both the type of association test and variant category that produced the lowest gene-trait *p* values were largely consistent in both analyses (Fig. 3).

**Clinical relevance of biomarker associations**

Genes identified by biomarker associations were highly enriched for those involved in Mendelian disorders (odds ratio 4.47, *p* value $3.08 \times 10^{-16}$, one-sided Fisher's exact test), as determined by querying the Online Mendelian Inheritance in Man database (OMIM, https://omim.org/). Seventy-four out of 125 (59.2%) distinct (i.e., non-overlapping) genes identified had at least one OMIM entry. Our results show that changes at the biomarker level are detectable even in the 27 genes for which only recessive disorders were listed.

We further examined whether rare variants in these genes were associated with binary or continuous traits at larger sample sizes by comparing with two studies published at the time of writing[15,16] (Supplementary Data 4). Twenty-one out of 125 genes associated with biomarkers were also associated with one or more binary traits, whereas 14 genes were associated with non-hematological quantitative traits in either study (Table 2).

Often such cases could be placed in a plausible causal context. For example, we found variants in *SLC22A12*, *ABCG2*, and *ALDH16A1* to be associated with increased levels of urate, a causal factor for gout, which they were also associated with. Another example are genes associated with the growth hormone IGF-1 (*IGFALS*, *GH1*, *GHRH*, and *PAPPA2*), which are also associated with anthropometric traits (e.g., sitting height). We identified patterns of biomarker associations for *GIGYF1* consistent with Type II diabetes (T2D), and an association with

T2D was confirmed at larger sample sizes[15,16] and other independent studies[31–33]. Six out of seven biomarker associations for *JAK2* were only identified by kernel-based tests, and the gain of function variant driving these associations (V617F or rs77375493[34]) was also associated with myeloproliferative neoplasms. In other cases, trait-biomarker relationships were not readily apparent, such as the association of variants in *SYNJ2* with lower levels of γ-glutamyltransferase and increased risk of hearing loss.

Separating variant effect categories during association testing and comparing kernel-based to gene-based collapsing tests allowed us to further interpret our results, as illustrated with several examples below.

**Combining variant annotations yields six associations for G6PC**

We found five associations for *G6PC*, three of which were only found when combining protein LOF and missense variants with gene-based variant collapsing tests (alkaline phosphatase, SHBG, and urate). Variants in *G6PC* cause Glucose-6-phosphatase deficiency type Ia[35,36], an autosomal recessive disease categorized by growth retardation, enlarged kidneys and liver, low blood glucose, high blood lipid, and uric acid levels. Consistent with signs of inflammation and impaired kidney and liver function, we found elevated levels of alkaline phosphatase ($p = 1.15 \times 10^{-9}$), γ-glutamyltransferase ($p = 2.64 \times 10^{-11}$), and C-reactive protein ($p = 1.62 \times 10^{-13}$) in individuals with predicted high-impact missense or pLOF mutations in *G6PC*. We further identified a significant association with decreased levels of sex hormone-binding globulin (SHBG, $p = 2.88 \times 10^{-9}$), which is primarily produced in the liver[37]. All *p* values above are those given by gene-based weighted variant collapsing tests combining missense and pLOF variants, which gave the lowest *p* values for all these associations. While glucose-6-

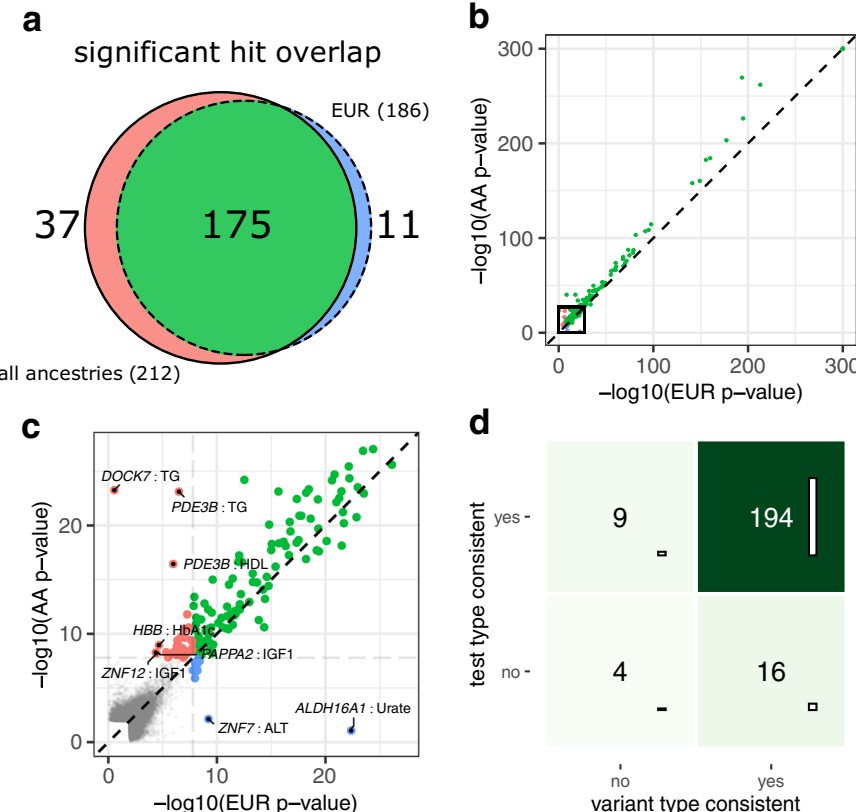

**Fig. 3 | Comparison of EUR vs. all-ancestry (AA) analyses. a** Venn diagram of the number of significant gene-biomarker associations ($p < 1.61 \times 10^{-8}$, FWER ≤ 0.05) identified in either analysis (Supplementary Data 1 and 3). **b** Scatter plot showing the smallest $p$ value across variant-effect- and test-types for each gene-biomarker association in the EUR-analysis (x-axis) vs. in the analysis with all ancestries (y-axis) on the negative log10-scale. The thick black box denotes the area highlighted in panel (**c**). Associations significant in either analysis are color-coded according to (**a**) and drawn thicker. The significance threshold is drawn as a light gray dashed horizontal/vertical line. Non-significant associations are drawn in gray. Associations with $p > 0.1$ in both analyses are not shown. **d** For significant gene-biomarker associations in either analysis, we show if the variant effect types and the test-types (gbvc or kernel-based) which gave the smallest $p$ values were consistent between the two analyses.

phosphatase deficiency type Ia is a rare recessive disease, our findings show that altered biomarker levels indicative of mild symptoms are detectable in heterozygous carriers of missense and LOF variants in the *G6PC* gene, a pattern which was consistent for many of the identified genes.

**Novel potential gain of function variants in *PIEZO1***

Our testing strategy allowed us to identify genes in which specific variant categories might play an important role. One such example was *PIEZO1*, a mechanosensitive cation channel[38], which we found associated with the diabetes marker HbA1c.

For weighted gene-based variant collapsing tests, we found a significant negative association of missense variants with HbA1c ($p = 7.47 \times 10^{-29}$), while the test for pLOF variants was not significant ($p = 0.89$, 609 carriers). By far the lowest $p$ value for this gene was given by the kernel-based test for missense variants ($p = 2.8 \times 10^{-108}$). The large differences between variant categories and types of association test lead us to closer investigate the 858 predicted high-impact missense variants in 7412 individuals for this gene.

We performed single-variant score tests and identified multiple missense variants with strong negative associations with HbA1c (Table 3). One of these variants, 16:88719665:G:A or T2127M (rs587776991) is a gain of function variant that slows down inactivation kinetics of PIEZO1 in patients with dehydrated hereditary stomatocytosis (a disorder of red blood cells), together with other gain of function variants[39–41]. Decreased levels of HbA1c had previously been observed in individuals with red blood cell disorders[42–44].

We therefore hypothesized that the other highly significant variants could also potentially be gain of function variants. We grouped the missense variants within *PIEZO1* by the amino acid positions they affected and performed local variant collapsing. This allowed us to identify other positions in *PIEZO1* (e.g., 2110R or 2474V) that are potentially sensitive to gain of function mutations (Table 3 and Fig. 4).

When comparing the results of our all-ancestry analysis, which includes ancestry-based variant pruning, to a version which did not, we came across another missense variant (L2277M, 16:88716656:G:T), which was much more common in individuals with inferred South Asian ancestry (SAS, $MAF_{SAS} = 3.45\%$) than European ancestry ($MAF_{EUR} = 0.006\%$, $p = 0$, two-sided Fisher's exact test), and strongly associated with decreased levels of HbA1c ($\beta = -0.71 \pm 0.06$ s.d., $p = 1.16 \times 10^{-29}$, score test restricted to individuals of inferred SAS ancestry, Supplementary Fig. 4). L2277M has been linked to dehydrated hereditary stomatocytosis[41]. This association remained significant when conditioning on a recently identified nearby non-coding variant (16:88784993:C:G, $p = 9.1 \times 10^{13}$)[45], yet, not the other way around ($p = 0.55$). 16:88784993:C:G could therefore be tagging signal from the potentially causal gain of function variant L2277M. We were able to replicate this association in individuals with extended EUR ancestry ($p = 2.73 \times 10^{-6}$, score test, Methods).

Consistent with the role of red blood cell disorders, we also found associations of *RHAG* (Rh Associated Glycoprotein) and *SPTA1* with decreased levels of HbA1c. Mutations in *RHAG* cause overhydrated hereditary stomatocytosis[46], while *SPTA1* mutations cause hereditary elliptocytosis[47].

**Table 2 | Significant genes with non-hematological trait associations in other RVAS**

| category | gene name | biomarker | analysis | p value | effect | test | $N_{variant}$ | $N_{carrier}$ | OMIM | traits (examples)[15,16] |
|---|---|---|---|---|---|---|---|---|---|---|
| bone and joint | HSPG2 | ALP | AA | 2.27e-19 | m | K | 1424 | 10,275 | r | repair of conjunctiva |
| bone and joint | B4GALNT3 | ALP | AA | 3.91e-14 | m | gbvc | 246 | 2499 | | height |
| bone and joint | FLG | Vitamin D | AA | 1.54e-10 | p | gbvc | 350 | 2448 | d,r | asthma, eczema |
| cardiovascular | APOB | LDL direct (+5) | AA | 9.69e-159 | m | K | 855 | 3948 | d,r | high cholesterol (diag.) |
| cardiovascular | PCSK9 | LDL direct (+2) | AA | 4.58e-88 | m | gbvc | 186 | 968 | d | high cholesterol (diag.) |
| cardiovascular | APOC3 | Triglycerides (+2) | AA | 2.18e-74 | m | K | 28 | 297 | m | lipid-lowering med |
| cardiovascular | LDLR | LDL direct (+3) | AA | 2.29e-50 | m | gbvc | 214 | 1303 | d,r | high cholesterol (diag.) |
| cardiovascular | PDE3B | Triglycerides (+1) | AA | 7.66e-24 | m | K | 225 | 1510 | | height, mass |
| cardiovascular | JAK2 | HDL (+6) | AA | 5.22e-19 | m | K | 254 | 1022 | m,s | neoplasms |
| cardiovascular | G6PC | CRP (+4) | AA | 1.62e-13 | m | gbvc | 85 | 828 | r | arm fat free mass |
| cardiovascular | TM6SF2 | Triglycerides (+2) | AA | 3.10e-13 | m | gbvc | 85 | 1392 | | liver fat percentage |
| cardiovascular | SRSF2 | HDL | EUR* | 6.19e-09 | m | K | 10 | 51 | | myeloid leukemia |
| diabetes | PIEZO1 | HbA1c | AA | 5.59e-108 | m | K | 1,011 | 8021 | d,r | varicose veins, height, mass |
| diabetes | GCK | HbA1c (+1) | EUR | 1.36e-40 | m | gbvc | 53 | 144 | d,r | T2D |
| diabetes | GIGYF1 | HbA1c (+4) | AA | 1.14e-10 | p | gbvc | 41 | 68 | | T2D, education score Engl. |
| diabetes | HBB | HbA1c | AA | 1.10e-09 | m | K | 33 | 92 | d,r | thalassemia |
| hormonal | SHBG | SHBG (+1) | AA | 4.64e-227 | m | K | 105 | 669 | | heel bone mineral density |
| hormonal | IGFALS | IGF-1 | AA | 1.14e-78 | m | K | 242 | 1900 | r | height, mass |
| hormonal | GH1 | IGF-1 | AA | 1.85e-10 | s | gbvc | 49 | 470 | d,r | height, mass |
| hormonal | GHRH | IGF-1 | AA | 2.07e-09 | m | gbvc | 22 | 117 | m | height, mass |
| hormonal | PAPPA2 | IGF-1 | AA | 5.38e-09 | m | gbvc | 288 | 1208 | r | height, mass |
| liver | UGT1A1 | Total bilirubin (+1) | AA | 2.90e-67 | m | K | 148 | 864 | m,r | Gilbert's syndrome |
| liver | ABCB4 | ALT | AA | 2.30e-10 | m | gbvc | 226 | 1192 | d,r | cholelithiasis |
| liver | SYNJ2 | GGT | AA | 1.55e-08 | m | gbvc | 434 | 3020 | | hearing loss |
| renal | SLC22A12 | Urate (+1) | AA | 0 | m | gbvc | 151 | 1044 | r | gout |
| renal | ALDH16A1 | Urate | EUR* | 4.40e-23 | m | K | 266 | 2060 | | gout |
| renal | ABCG2 | Urate | AA | 2.19e-14 | m | gbvc | 170 | 1575 | m | gout |
| renal | TNFRSF13B | Total protein | AA | 1.62e-12 | m | gbvc | 76 | 898 | d,r | interpol. age when op. |
| renal | ANKRD12 | Total protein | AA | 3.18e-11 | p | gbvc | 53 | 121 | | walking pace: slow |
| renal | PKD1 | Creatinine | EUR* | 2.77e-10 | m | K | 1,255 | 10,695 | d | cystic kidney disease |

Table showing the most significant gene-biomarker associations (FWER ≤ 0.05) for all genes that were also significantly associated with non-hematological traits (e.g., disease diagnoses or anthropometric traits) in ref. 15 or ref. 16 in either the all-ancestry analysis (AA) or the EUR-ancestry analysis (EUR). Only example traits are shown in column "traits". The full data are available in Supplementary Data 4. The numbers in brackets indicate how many other biomarkers are associated with the same gene in our analysis. effect: the lead variant effect type (m: missense pLOF, p: pLOF, s: splice + pLOF). test: the test-type which gave the lowest p value. OMIM: If disorders are linked to the gene in OMIM, their inheritance patterns (d: dominant, m: missing, r: recessive, s: somatic). Associations marked with (*) were not significant in the AA-analysis, see Supplementary Data 1 and 3 for details.

While it has been suggested that *PIEZO1* stimulates insulin release[48], the decreased levels of HbA1c we observed in individuals with *PIEZO1*-variants are more likely explained by (perhaps subclinical) forms of stomatocytosis or other abnormalities in red blood cells resulting from increased membrane permeability, i.e., a gain of function[49].

**Position-specific association of *ABCA1* variants with inflammation marker CRP**

We found four significant associations of *ABCA1* with biomarker levels. Three of these, namely the associations with Apolipoprotein A, HDL cholesterol, and cholesterol, are directly related to its role as an ATP-dependent transporter of cholesterol[50]. In line with previous findings, in our gene-based variant collapsing analysis we found both pLOF and high-impact missense variants to be strongly associated with decreased serum levels of these biomarkers[51].

Yet, one additional association with the inflammation marker C-reactive protein (CRP) was only identified by kernel-based association tests ($p = 1.33 \times 10^{-27}$). This prompted us to further investigate the 336 high-impact missense variants observed in *ABCA1*. Single-variant score tests and collapsing by amino acid position identified two missense variants (9:104831048:C:A, 9:104831048:C:G) in one of the extracellular domains affecting the same amino acid (W590), which were associated with strongly decreased levels of CRP (Fig. 4). The two

variants carried most of the signal in this gene with single-variant p values of $6.3 \times 10^{-32}$ for W590L (A allele, 54 carriers) and $8.09 \times 10^{-8}$ for W590S (G allele, 12 carriers, score test).

The W590S-variant leads to reduced cholesterol and phospholipid efflux, while retaining expression and ability to bind APOA1[52,53]. The other and more common variant, W590L, has been observed[54,55], but to our knowledge, not experimentally evaluated.

The binding of APOA1 to ABCA1 activates anti-inflammatory pathways via JAK2 and STAT3 in macrophages[56]. Because W590S has been shown to slow down dissociation of bound APOA1[53], this provides a plausible causal mechanism for the reduced levels of CRP we observe in carriers of the W590S-variant. We hypothesize that W590L might act through the same mechanism. This property could set these variants apart from other missense variants in *ABCA1*, which have been reported to abolish the binding of APOA1[52].

*ABCA1* could therefore be a gene in which some variants elicit both a gain of function (slower dissociation of APOA1) and a loss of function (decreased cholesterol efflux) with distinct effects on different biomarkers.

**Unique associations identified by splice predictions**

In total, we identified six gene-biomarker associations exclusively when incorporating SpliceAI variant effect predictions, of which four

**Table 3 | Potential *PIEZO1* gain of function variants**

| variant id | position | weight | variant | $N_{carrier}$ | variant p val. | $\beta_{variant}$ | ± | position p value | $\beta_{position}$ | ± |
|---|---|---|---|---|---|---|---|---|---|---|
| 16:88736318:C:T | 463 | 0.98 | A/T | 109 | 1.52e-10 | −0.56 | 0.09 | 2.01e-10 | −0.56 | 0.09 |
| 16:88736317:G:A | 463 | 1.00 | A/V | 1 | 8.97e-01 | 0.12 | 0.92 | 2.01e-10 | −0.56 | 0.09 |
| 16:88734895:G:A | 610 | 0.68 | L/F | 63 | 4.82e-08 | −0.63 | 0.12 | 9.55e-08 | −0.71 | 0.13 |
| 16:88734894:A:C | 610 | 0.99 | L/R | 4 | 7.00e-01 | −0.18 | 0.46 | 9.55e-08 | −0.71 | 0.13 |
| 16:88731880:C:G | 1008 | 0.99 | G/R | 65 | 7.20e-14 | −0.85 | 0.11 | 4.30e-15 | −0.88 | 0.11 |
| 16:88731880:C:T | 1008 | 0.99 | G/R | 3 | 1.18e-02 | −1.33 | 0.53 | 4.30e-15 | −0.88 | 0.11 |
| 16:88726891:A:C | 1175 | 1.00 | F/V | 4 | 4.71e-08 | −2.50 | 0.46 | 4.71e-08 | −2.50 | 0.46 |
| 16:88726565:C:T | 1260 | 0.71 | V/I | 78 | 8.03e-08 | −0.56 | 0.10 | 2.65e-08 | −0.68 | 0.12 |
| 16:88726565:C:A | 1260 | 1.00 | V/F | 1 | 6.67e-02 | −1.68 | 0.92 | 2.65e-08 | −0.68 | 0.12 |
| 16:88721626:C:G | 1772 | 0.99 | R/P | 12 | 3.11e-05 | −1.10 | 0.26 | 1.61e-08 | −1.07 | 0.19 |
| 16:88721627:G:C | 1772 | 0.98 | R/G | 10 | 9.12e-04 | −0.96 | 0.29 | 1.61e-08 | −1.07 | 0.19 |
| 16:88721626:C:A | 1772 | 0.98 | R/L | 1 | 3.90e-02 | −1.89 | 0.92 | 1.61e-08 | −1.07 | 0.19 |
| 16:88721627:G:A | 1772 | 1.00 | R/C | 1 | 4.89e-01 | −0.63 | 0.92 | 1.61e-08 | −1.07 | 0.19 |
| 16:88721423:C:G | 1804 | 0.87 | G/A | 31 | 7.92e-11 | −1.07 | 0.16 | 7.92e-11 | −1.15 | 0.18 |
| 16:88719717:G:A | 2110 | 1.00 | R/W | 9 | 6.68e-18 | −2.63 | 0.31 | 5.28e-33 | −2.66 | 0.22 |
| 16:88719716:C:T | 2110 | 0.89 | R/Q | 9 | 1.03e-16 | −2.53 | 0.31 | 5.28e-33 | −2.66 | 0.22 |
| 16:88719665:G:A | 2127 | 1.00 | T/M | 22 | 1.11e-31 | −2.29 | 0.20 | 1.11e-31 | −2.29 | 0.20 |
| 16:88716234:C:T | 2365 | 0.81 | G/R | 9 | 3.74e-16 | −2.49 | 0.31 | 3.74e-16 | −2.76 | 0.34 |
| 16:88715751:C:T | 2474 | 0.99 | V/M | 101 | 3.77e-08 | −0.50 | 0.09 | 9.58e-09 | −0.52 | 0.09 |
| 16:88715751:C:G | 2474 | 0.88 | V/L | 2 | 2.99e-02 | −1.41 | 0.65 | 9.58e-09 | −0.52 | 0.09 |
| 16:88715751:C:A | 2474 | 0.88 | V/L | 1 | 8.00e-01 | −0.23 | 0.92 | 9.58e-09 | −0.52 | 0.09 |

Variants are grouped and ordered by amino acid position and single-variant p values. All variants with position p values below $10^{-7}$ are shown. weight: impact score; $N_{carrier}$ number of carriers, variant p val.: single-variant p value (score test); $\beta_{variant}$: variant linear regression effect size (±standard error); position p value: p value when collapsing variants by position after weighting (score test); $\beta_{position}$: position linear regression effect size after weighting and variant collapsing (±standard error). Positions relate to the ENST00000301015 transcript.

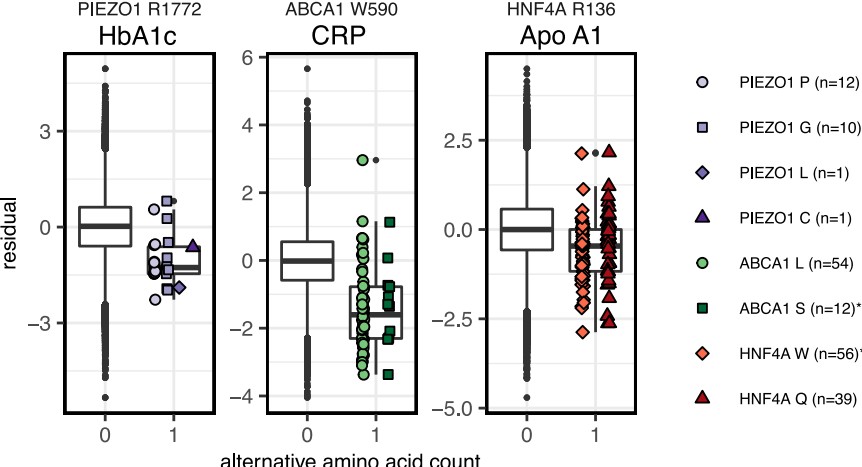

**Fig. 4 | Local collapsing of missense variants.** Dosage box plots showing the alternative amino acid counts (x-axis) against the covariate-adjusted quantile-transformed phenotypes (y-axis). Collapsing variants by amino acid position within significant genes (FWER ≤ 0.05) identified negative associations of *PIEZO1* R1772-variants with HbA1c ($p = 1.61 \times 10^{-8}$), *ABCA1* W590-variants with C-reactive protein ($p = 3.43 \times 10^{-38}$), and *HNF4A* R136-variants with Apolipoprotein A ($p = 1.2 \times 10^{-10}$). P values were derived using the score test after weighting and collapsing variants (Methods). Collapsing together with ClinVar variants with reported conditions (marked with "*") helps place novel variants into disease context. For all three associations, collapsed p values were lower than those of the single variants. Center lines denote the medians. The lower and upper hinges indicate 25th and 75th percentiles, whiskers extend to the largest/lowest values no further than $1.5 \times$ IQR away from the upper/lower hinges and black points denote outliers. Maxima and minima, from left to right: −5.33 to 4.95, −4.05 to 5.69, and −4.7 to 4.5.

were only found using kernel-based association tests. Specifically, we found associations of variants in *SLC9A5* with ApoA and HDL cholesterol, *NDUFB8* with Aspartate aminotransferase, *GH1* with IGF-1, *ECE1* with Alkaline phosphatase, and *KDM6B* with SHBG.

Most of these associations were mainly caused by single variants. An exception was the known association of *GH1* (growth hormone 1) with IGF-1[57], one of the two hits in this subset also found by gene-based

variant collapsing. The interpretation of single highly significant variants driving these associations could not necessarily be narrowed down to a single mechanism. For example, the predicted splice variant in the last exon responsible for the two significant associations of *SLC9A5* (16:67270978:G:A, 29 carriers, Fig. 5a) was also a missense variant (albeit with low to moderate predicted impact[55]). *ECE1* and *KDM6B* lie in proximity to the genes coding for the biomarkers they

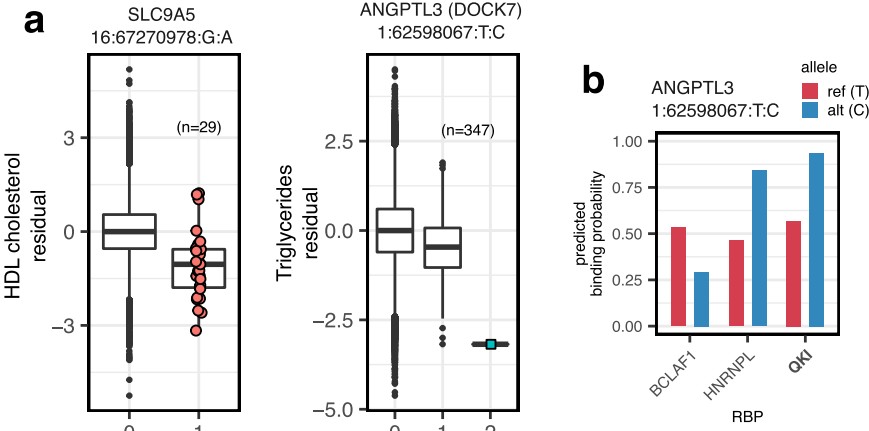

**Fig. 5 | Variants prioritized by deep learning models. a** Dosage box plots showing covariate-adjusted quantile-transformed phenotypes against minor allele counts for variants in *SLC9A5* and *ANGPTL3/DOCK7*. A predicted splice variant 16:67270978:G:A is negatively associated with HDL cholesterol ($p = 7.83 \times 10^{-12}$, score test), whereas intronic 1:62598067:T:C is negatively associated with Triglycerides ($p = 1.37 \times 10^{-25}$, score test). The numbers in brackets denote the number of carriers of at least one alternative allele. Center lines denote the medians. The lower and upper hinges indicate 25th and 75th percentiles. Whiskers extend to the largest/lowest values no further than 1.5 × IQR away from the upper/lower hinges and black points denote outliers. Maxima and minima, from left to right: −5.24 to 5.18, −4.62 to 4.5. **b** DeepRiPe binding probabilities for 1:62598067:T:C for three RBPs in HepG2 cells. While predicted probabilities for the reference sequence are ambiguous, the alternative allele shifts binding probabilities in favor of QKI and HNRNPL. All RBPs with absolute predicted variant effects above 0.2 and binding probabilities greater than 0.5 for either reference or alternative alleles are shown.

were found associated with (*ALPL* and *SHBG*), therefore we could not exclude transcriptional cis-regulatory effects as the cause of these associations.

## Associations identified by RBP-binding predictions

Out of the nine significant associations we identified using DeepRiPe variant effect predictions, the associations of *ANGPTL3* with Triglycerides, and *AGPAT4*, *PLG*, and *LPA* with Lipoprotein A had also been found using other models. The extreme heritability of Lipoprotein A, which is largely due to variation in the *LPA* gene[58,59], makes it hard to interpret the associations for the lead genes *LPA*, *AGPAT4*, and *PLG*, that all lie within a megabase distance to *LPA*. The association we observed for *ANGPTL3* was largely driven by a single intronic variant (1:62598067:T:C, Fig. 5), which was predicted to increase the binding probability of QKI. The same variant on the opposite strand was also responsible for the association of *DOCK7* with triglycerides. However, we determined that *ANGPTL3* was more likely the causal gene, given the associations of *ANGPTL3* with triglycerides we had independently found with other variant categories and the close proximity of an *ANGPTL3* exon.

We further investigated this variant by assessing the binding probabilities of RBPs beyond the six RBPs in focus here that are represented in DeepRiPe. We found that the variant was also predicted to decrease the binding probability of BCLAF1, a factor related to mRNA processing[60], and increase the binding of splice-regulator HNRNPL (Fig. 5). Using attribution maps[25], we found that instead of strengthening or inserting a QKI binding motif, this variant weakens a splice donor signal in the presence of upstream binding motifs for the splicing regulators QKI and HNRNPL (Supplementary Fig. 5). SpliceAI predicted only a weak upstream donor loss (0.02) for this variant, which was well below the threshold of 0.1 we used to identify splice-altering variants, but indicative of the same trend.

The remaining five associations exclusively identified by association tests incorporating RBP-binding predictions were those of *ATG16L1* with total bilirubin, *SOD2* with Lipoprotein A, *SLC39A4* with Alanine aminotransferase and *SHARPIN* with Alanine aminotransferase.

Both *SHARPIN* and *SLC39A4* lie within half a megabase of the Alanine aminotransferase gene (*GPT*), therefore we could not exclude potential transcriptional cis-regulatory effects as the true cause for

these associations. Furthermore, single variants carried most of the signal for both genes.

Common intronic variants of *ATG16L1* were previously found to be associated with Crohn's disease, inflammatory bowel disease[61,62], and increased bilirubin levels[23,63,64]. However, the association remained significant after conditioning on the intronic top variant from a previous study[23].

## Combined likelihood-ratio and score tests (sLRT)

In order to benefit from both the speed of the score test[8] and the higher power of the LRT in the presence of large effect sizes[11,12], we investigated the use of a combination of these tests, which we call the sLRT (score-LRT). The sLRT is a likelihood-ratio test that is performed only when initial score tests reach nominal significance at a given cutoff *t*. If this threshold is not reached, it returns the score test *p* value. Throughout our analysis, we used $t = 0.1$ and found that it was unlikely that a larger threshold would have identified many more associations (Supplementary Fig. 6). As the run time is dominated by the cost of computing the LRT, this test can achieve a computational speedup factor of roughly $1/t = 10$ over the LRT under the null hypothesis.

We found that the sLRT was able to identify more significant associations than the score test for missense and splice variants when performing kernel-based association tests (Supplementary Fig. 7). However, a large fraction of these additional associations (31% for splice variants and 48% for missense variants) could also be identified by gene-based variant collapsing for which the sLRT and score test gave almost identical results.

Nevertheless, we found the majority of the remaining additional associations to be plausible and/or previously reported in other association studies and therefore used the sLRT throughout our analysis, except for pLOF variants, where we only performed a gene-based variant collapsing test.

While the score test is the locally most powerful test (i.e., for alternatives that are close to the null hypothesis[65]), our empirical results suggest that the restricted LRT finds more associations. Based on simulations, ref. 11 found that the LRT has higher power if variant effect sizes are large, which is true for many rare variants observed in exome sequencing studies.

## Discussion

In our analysis, we combined gene-based variant collapsing and kernel-based tests under a common framework and performed functionally informed gene-based tests for rare variants with 30 biomarkers.

Overall, our approach was successful at identifying disease genes without explicitly using disease diagnoses themselves, even for recessive diseases (e.g., *G6PC* and glycogen storage disease Ia[35] or *ABCA1* and Tangier disease[66]), or diseases with mixed inheritance patterns, while keeping the number of tests low compared to phenome-wide association studies.

While some of the changes in biomarker levels we detected might be be subclinical, they could interfere with the diagnosis of common conditions which rely on biomarkers. To prevent misdiagnoses and enable preventative care, our results could aid the design of targeted sequencing panels that focus on the genes with the highest impacts. For example, we found 8% of the participants to harbor at least one protein LOF variant in any of the significant loci for that variant category. In other words, it is fairly common to have at least one uncommon LOF variant with potential effects on biomarker levels. Genotyping the *PIEZO1*-L2277M variant in individuals with South Asian ancestry could improve diagnostics in that population, especially if patients present with symptoms typical of dehydrated stomatocytosis, e.g., hemolysis or hyperferritinemia[41].

One major contribution of our study was the design and successful application of kernel-based tests that incorporate quantitative functional variant effect predictions for large exome sequencing data. A previous study had tried to apply kernel-based score tests using SKAT[8] on the 50k WES release, but concluded that results were difficult to reproduce[13]. In contrast to their approach, we did not re-weigh variants according to allele frequencies. Furthermore, we showed that a computationally efficient combination of the restricted LRT and score test has potentially higher power than the score test alone (possibly due to the large effect sizes) and identified more associations than gene-based variant collapsing tests.

Ancestry-based variant pruning was necessary to prevent inflation of test statistics for certain biomarkers. This approach worked for us in practice, but could lose power if causal variants are discarded. Future studies could focus on integrating the methods we propose with more sophisticated ways to account for confounding that allow for larger allele frequency differences between individuals of different genetic ancestry[11,67,68].

When comparing gene-based collapsing and kernel-based tests for missense variants, we found kernel-based tests to have advantages in the presence of gain of function variants (*PIEZO1*, *ABCA1*, and *JAK2*), where they identified plausible causal associations missed by gene-based collapsing tests. These genes likely are examples of a low fraction of causal variants, a regime in which kernel-based tests are statistically more powerful than gene-based collapsing[7].

While we found a large overlap between our associations and those found in other studies[15,16,23], the differences highlight the sensitivity of gene-based tests to qualifying criteria for rare variants (i.e., allele frequencies or variant impact predictions), which can make them harder to reproduce (Supplementary Data 4). By making our analysis pipeline public, we hope to increase reproducibility and enable others to explore different qualifying criteria more easily.

We demonstrated how local collapsing of missense variants by amino acid position aids interpretation and causal reasoning in the presence of previously validated variants. Local collapsing was directly built into the kernel-based tests we performed for missense variants, where it affected 20% of variants, a number which will increase with larger datasets.

We explored the use of deep learning-derived variant effect predictions for splicing and the binding of RBPs. The restriction to exon-proximal regions meant we only observed a fraction of the variants potentially acting through these mechanisms. Associations found by incorporating splice predictions largely overlapped with those identified with pLOF variants (which included simple splice donor/acceptor variants). While we found some associations exclusively with splice predictions, these were mostly due to single variants and would need further validation (e.g., *SLC9A5*). Similar reasoning holds for the associations found with predictions for RBP-binding.

We anticipate that deep learning-based predictions will become more valuable for non-coding regions in whole-genome sequencing studies, for which the approaches we developed will also be applicable. Deep learning-derived functional annotations have been considered in other studies in the context of association testing. Proposed methods include signed LD-score regression[69], or the association tests presented in DeepWAS[70]. However, these methods have not been designed for very rare variants.

In future studies, methods like AlphaFold[71] could allow specific testing of effects on protein folding. Methods that allow predicting residue-residue interactions within proteins could enable the mostly unsupervised identification of protein domains and their separate testing[72]. The methodological advances and practices in this association study also apply to those situations and serve as potential baselines for functionally informed kernel-based association tests with rare variants.

## Methods

### Ethics statement

UK Biobank protocols are overseen by the UK Biobank Ethics Advisory Committee (EAC). Informed consent was obtained for all participants. Participants that revoked consent were removed from the analysis. The original approval for the UK Biobank was granted in 2011 by the National Research Ethics Service (NRES) Committee North West - Haydock. The approval was renewed in 2016 and 2021 by the Health Research Authority, North West - Haydock Research Ethics Committee. This research has been conducted using the UK Biobank resource under application number 40502.

### UK biobank data processing

All 30 blood biochemistry biomarkers (category 17518) from the UK Biobank were quantile-transformed to match a normal distribution with mean 0 and unit standard deviation using scikit-learn (v0.22.2)[73]. For testosterone, which showed a clear bimodal distribution based on sex, quantile transformation was performed separately for both sexes. We performed ancestry scoring as described in ref. [74] based on the 1000 Genomes super populations[75], which was used to prune variants and select participants with similar genetic ancestry.

Sex, BMI, age at recruitment, smoking status, genetic principal components, and continuous ancestry predictions were used as covariates (Supplementary Data 2). Smoking status (never, previous, current) was encoded in three separate binary variables. Participants with any missing covariates were excluded. We used the ukb_gen_samples_to_remove function of the ukbtools R-package (v0.11.3)[76] together with pre-computed relatedness scores (ukbA_rel_sP.txt, see UK Biobank Resource 531) to remove closely related individuals, keeping only one representative of groups that are related to the 3rd degree or less. After removing 6,293 related individuals and restricting to those with no missing covariates, 191,971 participants remained. This sample was 55% female (45% male) and the average age at recruitment was 56.5 years ($\sigma = 8$). Furthermore, the average BMI was 27.37 ($\sigma = 4.76$) and our subset contained 18,529 current and 66,988 previous smokers.

In our analysis, we made use of the PLINK-formatted exome sequencing genotype data. The final results presented in this manuscript were derived from the 200k WES release produced by the OQFE pipeline[21]. The UK Biobank pipeline already implements quality filters[21,77]. Additionally, we removed all variants that violated the Hardy–Weinberg equilibrium (HWE) assumption (HWE exact test *p*

value below the threshold of $10^{-5}$) and variants genotyped in less than 90% of participants.

We calculated minor allele frequencies within all unrelated participants with complete covariates (see above), and excluded variants with a study-wide allele frequency above 0.1%. In the analysis including all ancestries, we additionally performed ancestry-based pruning to remove variants with large allele frequency differences between ancestries. This lead to the exclusion of 1,033,382 variants. We found this step to be critical for preventing test statistic inflation for kernel-based tests with certain phenotypes. We did not analyze variants on sex chromosomes.

In the all-ancestry analysis 15,701,695 variants passed these filters, of which 45.87% were singletons. 12,793,493 were considered for the EUR ancestry analysis (42.2% singletons). We directly use UK Biobank variant identifiers (which include chromosome and 1-based hg38 positions) to name variants in order to facilitate comparisons.

## Variant effect prediction and annotation

**Protein loss of function and missense.** We predicted effects for all genetic variants that passed basic filtering using the Ensembl Variant Effect Predictor[17] (VEP, v101; cache version 97), including scores from Polyphen-2[18] (v2.2.2) and SIFT[19] (v5.2.2). All variants marked as splice_acceptor_variant, splice_donor_variant, frameshift_variant, stop_gained, stop_lost, or start_lost were considered protein loss of function (pLOF) variants as in ref. 13. We further annotated missense variants by calculating impact scores (averages between deleterious-probabilities given by PolyPhen-2 and SIFT), which were used to filter and weigh variants in the association tests. Specifically, Missense variants were included if their impact score was at least 0.8 or if they affected amino acid positions for which another variant with an impact score of at least 0.8 was observed.

**Splicing.** We retrieved published pre-computed variant effect predictions produced by the SpliceAI deep learning model[24] for single nucleotide polymorphisms. SpliceAI predicts the consequences of genetic variants for nearby splice sites, specifically splice donor loss/gain or splice acceptor loss/gain. We used the splice-site-proximal masked delta scores (v1.3). In the masked files, scores corresponding to the strengthening of annotated splice sites and weakening of non-annotated splice sites are set to 0, as these are generally less pathogenic. We included splice variants in the association tests if at least one of the four SpliceAI delta scores was greater or equal to 0.1. The maxima over the different delta scores for every variant were used to weigh variants in the association tests (details below).

**RBP-binding.** We predicted the effects of all genetic variants on the binding of 6 RNA-binding proteins (RBPs) using a modified version of the DeepRiPe deep neural network[25] in which predictions are purely sequence-based (implemented in keras with tensorflow backend, see Data availability statement). We predicted the differences in binding by subtracting the predictions for the reference alleles from those for the alternative allele[78], and used these variant effect predictions to filter and weigh variants during the association tests (details below). Variants were included into the association tests if at least one predicted effect on any of the RBPs had an absolute value greater or equal to 0.25.

## Ancestry scoring and ancestry-based variant pruning

We used GenoPred[74] to perform ancestry scoring of all UK Biobank participants. GenoPred uses an elastic net model based on the first 100 genetic principal components of the 1000 Genomes genetic data and super population assignments[79] to predict the genetic ancestry of individuals.

In the analysis, including all ancestries, we performed ancestry-based variant pruning, as follows. We limited the analysis to

participants with complete covariates and to variants with a study-wide MAF below 0.1% within those individuals. We then defined groups of individuals based on the ancestry prediction model. An individual was assigned to one of the five ancestry groups defined in the 1000 Genomes reference if the predicted probability for that ancestry was greater than 0.5. 182,288 participants were identified as having predominantly European ancestry (EUR), 4302 South Asian ancestry (SAS), 3775 African ancestry (AFR), 1126 East Asian ancestry (EAS), and 308 admixed American ancestry (AMR). About 172 individuals could not be placed in any of these groups, as all probabilities were below 0.5. These predictions were used to group individuals with similar genetic ancestry according to the 1000 Genomes and do not reflect ethnicity.

We then performed two-sided Fisher's exact tests to identify variants with large deviations from the study-wide EUR allele frequency in any of the other super populations and excluded variants with $p < 10^{-5}$ from the analysis. Variants selectively missing in any of the super populations were excluded.

The predictions from the ancestry prediction model were also used to define the group of participants which were used for the EUR-only analysis, where we applied a more stringent cutoff of $Pr(\text{EUR}) > 0.995$.

## Statistical models and tests

Let $N(\boldsymbol{\mu}; \boldsymbol{\Sigma})$ denote a multivariate Normal distribution with means $\boldsymbol{\mu}$ and a variance-covariance matrix $\boldsymbol{\Sigma}$. We wish to jointly test the association of $m$ genetic variants with a quantitative trait $\mathbf{y}$ for a sample of $N$ observations (i.e., participants) while controlling for $q$ covariates. Within the linear mixed model framework, $\mathbf{y}$ can be modeled as follows[8,9]:

$$\mathbf{y} \sim N(\mathbf{X}\boldsymbol{\alpha}; \sigma_e^2 \mathbf{I}_N + \sigma_g^2 \mathbf{K}_g), \tag{1}$$

where $\mathbf{X}$ is the $N \times q$ covariate design matrix (fixed effect) and $\boldsymbol{\alpha}$ is the vector of fixed-effect parameters, which together determine the mean values of $\mathbf{y}$. The variance-covariance matrix of $\mathbf{y}$ is composed of the independently distributed residual variance ($\mathbf{I}_N$ scaled by $\sigma_e^2$) and the kernel-matrix $\mathbf{K}_g$ (scaled by $\sigma_g^2$), which captures the genetic similarity between individuals. $\mathbf{K}_g$ is a function of the $N \times m$ matrix of mean-centered minor allele counts $\mathbf{G}$ (random effect) of the genetic variants we wish to test.

Any valid variance-covariance matrix can be substituted for $\mathbf{K}_g$. In order to use efficient algorithms for estimating the parameters $\sigma_e^2$ and $\sigma_g^2$ and performing association tests, we require $\mathbf{K}_g$ to be factored as a quadratic form[9,11]:

$$\mathbf{K}_g = \phi(\mathbf{G})\phi(\mathbf{G})^T, \tag{2}$$

where the function $\phi$ transforms $\mathbf{G}$ into intermediate variables before performing the test. Finding an appropriate function $\phi$ depends on the underlying biological assumptions and the available prior information. Gene-based variant collapsing approaches are a special case, in which the function $\phi$ returns an $N \times 1$ vector (a single variable) as output. Therefore kernel-based tests and variant collapsing methods can be treated under the same statistical framework. In our analysis, $\phi$ is a function that transforms $\mathbf{G}$ taking variant effect predictions and, for missense and RBP-variants, variant positions into account.

Regardless of the choice of kernel (and hence $\phi$) the statistical test is defined by the null hypothesis $H_0 : \sigma_g^2 = 0$, and the alternative hypothesis $H_1 : \sigma_g^2 > 0$ (i.e., a one-sided test). Both a score test and likelihood-ratio test (LRT) have been described for this application. While the score test is often chosen in statistical genetics applications due to its speed and software availability, the LRT has been shown to have higher power when effect sizes are large but is computationally more demanding[8,11,12].

In order to avoid computing the LRT for all genes but still profit from a potentially higher power, we performed score tests genome-wide and only performed the restricted LRT if score tests (within the specific variant category) reached nominal significance, an approach which we call the score-LRT (sLRT, see sections below). The sLRT returns the $p$ value for the score test if nominal significance was not reached, otherwise it returns the $p$ value for the likelihood-ratio test.

We sampled RLRT test statistics using fast exact algorithms described in ref. 80, and fit parametric null distributions to the pooled test statistics across genes in order to calculate $p$ values[11]. Null distribution parameters are available in Supplementary Data 1. We did this separately for different variant-effect and test-types. This method gave highly similar $p$ values to using gene-specific null distributions (Supplementary Figs. 9–12), but is faster as it requires fewer test statistics to be simulated per gene and fewer distributions to be fit. Furthermore, using the pooled distribution does not change the bound of the FWER compared to using gene-specific null distributions to calculate $p$ values (see sections below).

We applied the statistical framework above to perform both gene-based variants collapsing tests and kernel-based association tests, corresponding to different functions $\phi$ as detailed below. Additionally, for splice and missense variants, tests using only those variant categories and tests combining these variant categories with pLOF variants were integrated into single tests using the Cauchy combination test[28]. We adjust $p$ values for the total number of 3,091,910 tests in the all-ancestry analysis using Bonferroni correction (FWER ≤ 0.05), which lead to a cutoff of $1.6171 \times 10^{-8}$.

## Variant weight calculation

All association tests we performed incorporated variant weights, which were derived from variant effect predictions. All variant weights we used are numbers between 0 and 1. For protein LOF variants, all weights were set to 1. For missense variants, we calculated the weights as follows:

$$w_i = \frac{(1 - s_{i,\text{SIFT}}) + s_{i,\text{Polyphen}}}{2} \tag{3}$$

where $w_i$ is the weight for variant $i$. $s_{i,\text{SIFT}}$ and $s_{i,\text{Polyphen}}$ denotes the SIFT and Polyphen scores for variant $i$, respectively (potentially averaged across different transcript variants). This score can be interpreted as the average of the predicted probability of the variant being deleterious predicted by the two methods.

For splice variants, the weight $w_i$ for specific variant $i$, was set to the maximum of its four SpliceAI delta scores.

Regarding the predictions for the binding of RBPs, we proceeded as follows: While the experiments for the RBP QKI had been replicated in three cell lines, those for the other 5 RBPs had only been performed in a single cell line. As every replicate is a separate model output, this resulted in a total of eight predictions for every genetic variant. We predicted the binding probability of each RBP to sequences centered on the major and minor alleles, while applying 4 bp shifts around the center. We averaged four predictions across these small shifts to reduce variability. Finally, we calculated variant effect predictions $v_{ij}$ for each variant $i$ and RBP-replicate $j$ by subtracting the prediction for the reference allele ($p_{ij,\text{ref}}$) from the prediction for the alternative allele ($p_{ij,\text{alt}}$)[78]:

$$v_{ij} = p_{ij,\text{alt}} - p_{ij,\text{ref}} \tag{4}$$

These variant effect predictions are numbers between −1 and 1, where the sign denotes a gain of binding (+) or a loss of binding (−). They were used to determine variant weights and variant similarities during association testing (see below), where we set the weight $w_i$ of variant $i$ to the largest absolute value of $\mathbf{v}_i$.

## Gene-based variant collapsing tests

In gene-based variant collapsing, all qualifying variants overlapping a specific gene are collapsed into a single variable prior to association testing, i.e., $\phi(\mathbf{G})$ in (2) returns an $N \times 1$-vector. We modified the approach in ref. 13 by incorporating variant effect predictions as weights. Within a specific gene, any participant could carry 0 or more qualifying variants, where each variant $i$ has a weight $w_i$ (derived from variant effect prediction, see above). Specifically, the collapsed score is the largest weight of any of the variants observed for a specific participant, or 0 if no qualifying variants were observed for that participant. This score makes three assumptions: additive effects are negligible (or unrealistic), variants with larger weights dominate over those with smaller weights and all variants affect the quantitative trait in the same direction.

## Functionally informed kernel-based tests

The kernels we used in this analysis follow the general form:

$$\mathbf{K}_g = \mathbf{GWSWG}^T, \tag{5}$$

where $\mathbf{W}$ is an $m \times m$ diagonal matrix containing the square roots of variant weights on the diagonal and the $m \times m$ matrix $\mathbf{S}$ captures similarities between the genetic variants. $\mathbf{G}$ is the $n \times m$ matrix of mean-centered minor allele counts of the qualifying variants within the gene to be tested. $\mathbf{S}$ can be interpreted as the variance-covariance matrix of regression coefficients of intermediate variables $\mathbf{GW}$. We use $\mathbf{W}$ and $\mathbf{S}$ to incorporate variant effect predictions (and other variant annotations) into the association tests.

While a shared regression coefficient ($\mathbf{S} = \mathbf{1}_m \mathbf{1}_m^T$) might be a poor assumption in some cases, so can completely independent regression coefficients ($\mathbf{S} = \mathbf{I}_m$). The former, when substituted into (5), has been referred to as the weighted counting burden test, whereas the latter is commonly called the weighted linear kernel[81]. In our analysis, we define $\mathbf{S}$ based on available prior knowledge and type of variant effect prediction.

**Missense**. For the analysis of missense variants, we introduce the locally collapsing kernel. Local collapsing aggregates groups of variants into single variables before performing the association test. "Local" refers to the fact that the groups are defined by the proximity of variants in the DNA-, RNA-, or amino acid sequence. We grouped variants if they affect the same exact amino acid position of a specific gene. Once the groups are defined, local collapsing can be expressed as a matrix multiplication: $\mathbf{S} = \mathbf{CC}^T$ and the kernel (5) becomes:

$$\mathbf{K}_g = \mathbf{GWCC}^T\mathbf{WG}^T \tag{6}$$

Here $\mathbf{C}$ is the $m$-variants by $g$-groups collapsing matrix. Therefore $\mathbf{GWC}$ is the $n \times g$ weighted locally collapsed genotype matrix (where the columns now represent amino acid positions instead of single genetic variants). The columns of $\mathbf{C}$ define the group assignments and directionality of variant effects. For every variant $i$ from 1 to $m$ with (potentially signed) variant effect $v_i$ and group $j$ from 1 to $g$, $c_{ig} = \text{sgn}\, v_i$ if variant $i$ belongs to group $j$, else $c_{ig} = 0$. In our case, variant effect predictions were unsigned (all positive). The assumptions of the locally collapsing kernel are that variants within groups share a common regression coefficient once they have been scaled by and aligned with the direction of their variant effect predictions.

**RBP-binding**. Sometimes there are no clearly defined groups of variants or multiple (potentially directional) variant effect predictions need to be accounted for at once, and therefore, variants can't easily be collapsed. Given what we know about the location of variants and their predicted effects, we might still make assumptions about $\mathbf{S}$. As long as $\mathbf{S}$ is positive definite, we can find a suitable square root $\mathbf{L}$ so that $\mathbf{LL}^T = \mathbf{S}$ using the Cholesky decomposition. In the association tests involving directional predictions for the binding of RNA-binding proteins we calculated $\mathbf{S}$ by

forming the element-wise product of two $m \times m$ matrices:

$$\mathbf{S} = \mathbf{L}\mathbf{L}^T = \mathbf{Q} \circ \mathbf{R} \qquad (7)$$

Where $\mathbf{Q}$ captures the similarity of variants based on their variant effect predictions and $\mathbf{R}$ captures the similarity of variants based on their positions. Specifically, let $\mathbf{v}_i$ be the vector of variant effect predictions for variant $i$. Then the element $q_{ij}$ of $\mathbf{Q}$ is the cosine similarity between $\mathbf{v}_i$ and $\mathbf{v}_j$. We chose to model the position-dependent similarity with a Gaussian kernel. If $x_i$ is the chromosomal position of variant $i$, $r_{i,j} = \exp(-\gamma(x_i - x_j)^2)$, where we set $\gamma = -\frac{\log(0.5)}{50^2}$. At this value of $\gamma$ two variants that are 50 bp apart have a similarity of 0.5, which decays rapidly as the distance increases. As both $\mathbf{Q}$ and $\mathbf{R}$ are positive definite matrices, so is $\mathbf{Q} \circ \mathbf{R}$. This kernel makes the assumption that variants that are in close proximity and have aligned variant effect predictions should affect the phenotype in the same direction.

### Score- and likelihood-ratio test implementation

In order to use efficient algorithms for estimating the parameters $\sigma_e^2$ and $\sigma_g^2$ in (1) and performing association tests, we require $\mathbf{K}_g$ to be factored as a quadratic form as shown in (2)[9,11]. The function $\phi$ in (2) transforms the genotype matrix $\mathbf{G}$ into intermediate variables before performing the test.

The test statistic of the score test approximates the change of the log-likelihood of a model when including $\mathbf{K}_g$ over the null model, which does not include $\mathbf{K}_g$ ($\sigma_g^2 = 0$)[8]. We calculated test statistics using fast algorithms described in[11] and applied Davies' method for the calculation of $p$ values[82] with accuracy of $10^{-7}$ and $10^6$ iterations. Where Davies' method returned $p$ values of 0, or in the rare cases where Davies' method returned invalid (negative) $p$ values, we used saddle point approximation instead[83].

The test statistic of the restricted likelihood-ratio test is twice the difference between the log restricted likelihood of the alternative model and the null model[9]. We used FaST-LMM's LMM class[84] to fit the null and alternative models using restricted maximum likelihood and then calculated test statistics. To generate a null distribution we sampled 100 test statistics for every LR test, using our own port of RLRsim[80] in Python (as part of the seak package, see Code availability statement). Finally, we fit a parametric null distribution $\pi\chi_0^2 + (1 - \pi)a\chi_d^2$ with free parameters $\pi$, $a$, and $d$ to the pooled simulated test statistics using log-quantile regression on the 10% of largest test statistics[11], and used this distribution to calculate $p$ values as described in ref. 9 (Supplementary Data 1). We used separate null distributions for all variant-type to test-type and phenotype combinations (see details below).

We compared this approach to using gene-specific null distributions (i.e., fitting a separate null distribution for every test and gene, similar to the method described in ref. 12), and found that they produced highly similar results (Supplementary Figs. 9–11).

To this end, we performed two analyses: First, we looked at genes close to or below the genome-wide significance threshold. We compared the Pearson correlation of the log10 $p$ values derived from the genome-wide pooled or gene-specific null distributions for genes with association $p$ values below $10^{-6.5}$ in any of the initially performed tests. We did this separately for the different variant categories and test-types based on 250,000 gene-specific samples from the null distribution. Second, for every phenotype and variant category, we randomly sampled 100 genes with $p$ values above $10^{-6.5}$ (in any of the previously performed tests for that variant category) and cumulative minor allele counts of at least 5, and repeated the comparison (again based on 250,000 samples per test).

For associations close to or below the significance threshold, the average $r^2$ was 0.999 for kernel-based tests and 0.999 for gbvc tests. For non-significant associations, average $r^2$ values were 0.9897 for kernel-based tests, and 0.999 for gbvc tests. We conclude that the pooled null distribution is a good approximation of the individual gene-level null distributions. An example illustrating this approach is shown in Supplementary Fig. 12.

### Gene-based testing procedure summary

We performed gene-based tests for all protein-coding genes in the Ensembl 97 release. For all pLOF variants we performed gene-based variant collapsing using the score test genome-wide.

For missense variants, we performed both weighted gene-based variant collapsing and kernel-based association tests using the sLRT. For the kernel-based tests with missense variants, we designed a kernel that collapses variants by amino acid position (local collapsing) and weighs them by their impact score. Additionally, in cases where either missense-variant score test used in the sLRT was nominally significant ($p < 0.1$), we combined missense and protein LOF variants for joint tests. For these joint tests, we investigated both the use of joint weighted gene-based variant collapsing and a kernel-based test that combines collapsing of pLOF variants with local collapsing of missense variants by concatenation (detailed description below). The $p$ values of the combined tests were integrated using the Cauchy combination method[28] (individual $p$ values are reported in Supplementary Data 1).

For predicted splice variants, we followed the same strategy as for missense variants, however, we used the weighted linear kernel[8] without local collapsing instead. Finally, in the association tests, including variants predicted to change the binding of RBPs, we only performed kernel-based association tests using the sLRT. For this purpose, we designed a kernel that can take into account both variant positions and the direction of variant effects (as described above).

Because some of the genes in the Ensembl 97 release share exons, we encountered cases in which these genes shared associations caused by the same variants. We do not report these as distinct genes in the main text or abstract, but include the full list of 212 associations in Supplementary Data 1.

### sLRT detailed description

**Missense.** For missense variants, we iterated over all genes and performed score tests using gene-based variant collapsing and kernel-based tests (locally collapsing kernel), i.e., the diagonal elements $w_{ii}$ of $\mathbf{W}$ in (5) contained the square roots of the impact scores of variants. If either score test $p$ value was nominally significant ($p < 0.1$) we also performed the following steps: (1) Calculation of restricted likelihood-ratio test statistics (sLRT), (2) gene-based variant collapsing combining both missense and loss of function variants in a joint test, (3) concatenation of the collapsed pLOF variable to the locally collapsed weighted matrix of missense variant minor allele counts (**GWC**, Equation (6)) and a joint kernel-based RLRT. For all likelihood-ratio tests, we simulate 100 gene-specific test statistics from the null distribution each.

Once all genes were processed for a specific phenotype, we fit separate null distributions to the pooled simulated test statistics for each of the four groups of likelihood-ratio tests: collapsed missense variants (gbvc), jointly collapsed pLOF and missense variants (gbvc), locally collapsed missense variants (kernel-based), locally collapsed missense variants concatenated with collapsed pLOF variants (kernel-based). $P$ values were then calculated for all tests based on those distributions. Finally, the $p$ values for the two kernel-based tests, and the two gbvc tests were combined using the Cauchy combination test, resulting in a single kernel-based and a single gbvc test per gene.

We used the locally collapsing kernel in the kernel-based association tests for missense variants as it had given more unique associations and overall slightly lower $p$ values for the most significant genes in initial experiments on the 50k WES release, and was more interpretable compared to other approaches.

**Splicing.** For splice variants, we performed score tests using gene-based variant collapsing and the linear weighted kernel for all genes. Again, if either of the two score tests were nominally significant ($p < 0.1$), we performed likelihood-ratio tests (sLRT). As we did for missense variants, we then also performed combined association tests with protein loss of function variants using both gene-based variant collapsing and a kernel-based LRT. For the kernel-based test, we concatenated the protein LOF indicator variable to the matrix of weighted minor allele counts **GW** (Equation (5), where $\mathbf{S} = \mathbf{I_m}$). In the cases where a variant was annotated both as a splice variant and pLOF variant, we treated it as a pLOF variant in the joint tests. As we did for missense variants, after calculating $p$ values using four separate null distributions for every phenotype, we combined the two kernel-based tests and the two gene-based collapsing tests into single tests using the Cauchy combination test.

**RBP-binding.** For variants predicted to alter the binding of RBPs we only performed kernel-based association tests using the kernel in (5), where we used the largest absolute value of the variant effect predictions as the weights and calculated **S** as described above in (7). We iterated over all genes and performed gene-based score tests. Because the DeepRiPe variant effect predictions are strand-specific, we did this independently for genes on the forward or reverse strands. If the score test for a specific gene was nominally significant ($p < 0.1$), we performed the likelihood-ratio test for that gene (sLRT). If the variants tested also included variants annotated as protein loss of function variants, we removed them and repeated the tests to avoid false positives.

### Conditional association tests

For significant associations after multiple testing correction, we performed conditional association tests. For every significant gene-biomarker association, we identified single variants significantly associated with the same biomarker within ± 500kb of the gene start position, based on the summary statistics provided by ref. 23. We then conditioned on the single variant with the lowest $p$ value (if any) by incorporating it as a covariate in a gene-specific null model (in case of ties, the closest variant to the gene start position was chosen). We then fit the alternative model, calculated the model likelihoods and RLRT test statistics, and simulated 250,000 gene-specific RLRT test statistics for every alternative model (i.e., combinations of variant- and test-types). We fit parametric null distributions to these test statistics using the 10% of largest test statistics (as described above), and calculated $p$ values based on these null distributions. We then combined the $p$ values using the CCT (if combined tests with pLOF variants were performed). Conditional $p$ values and the variants that were conditioned are reported in Supplementary Data 1.

### Cross-referencing against GWAS databases

We queried the NHGRI-EBI GWAS Catalog[2] and PhenoScanner[26,27] in order to see if single variants within the genes we found significantly associated with a specific biomarker had already been reported to be associated with that biomarker. For each gene, we submitted region queries using the gene boundaries with the gwasrapidd[85] (v0.99.11) and phenoscanner (v1.0) R-packages. For PhenoScanner, we set the $p$ value threshold to $10^{-7}$. Matching our results to those contained in these databases required us to define a mapping of UK Biobank biomarkers to the Experimental Factor Ontology (EFO) terms used in those databases. This mapping is provided in Supplementary Data 2. Additionally, as EFO terms for PhenoScanner were not always defined, we performed the following matching: "Apolipoprotein B" (UKB phenotype) to "APOB apolipoprotein B" (PhenoScanner trait), "Cystatin C" to PhenoScanner traits "log eGFR cystatin C", "Serum cystatin c estimated glomerular filtration rate eGFR", and "Cystatin C in serum", and "Urea" to "Renal function related traits urea".

### PIEZO1-L2277M association tests

We used the ancestry classifications described above to define a group of individuals of SAS ancestry, and one of extended EUR ancestry (both using a cutoff of >0.5 in the ancestry classification model). This is a less stringent cutoff than that used in the EUR-analysis for all biomarkers and increased the number of observed carriers in the EUR-ancestry group from 5 to 21, all heterozygous. We used the same covariates as in the all-ancestry analysis.

Genotypes were derived from exome sequencing and we performed association tests using the score test with ancestry-specific null-models. For the association tests in the SAS group, we performed conditional tests by conditioning the test statistic on the genotypes of the 16:88784993:C:G variant, which were also derived from exome sequencing.

### FWER control for the pooled null distribution

For the RLRT we derive $p$ values from pooled gene-specific test statistics under the null hypothesis and use Bonferroni correction on these $p$ values to bound the FWER. Below we show why this approach does not change the bound of the FWER compared to using gene-specific null distributions to calculate $p$ values, assuming the pooled distribution is well estimated.

Let $I_i$ be a sample from a random variable. For $i$ from 1 to $n$, $I_i$ corresponds to the test statistics for gene $i$ under the null hypothesis. For a specific value of the test statistic $x$, the distribution function $F_i$ returns the $p$ value $p_i$:

$$F_i(x) = \Pr(I_i \le x) = p_i \qquad (8)$$

Let $M$ be the random variable arising from a uniform mixture of all $I_n$, i.e., $M = \bigcup_{i=1}^{n} I_i$, with corresponding distribution function:

$$F_m(x) = \Pr(M \le x) = p_m = \sum_{i=1}^{n} \frac{F_i(x)}{n} = \sum_{i=1}^{n} \frac{\Pr(I_i \le x)}{n} = \bar{p}_i \qquad (9)$$

I.e., when the mixture components are sampled to the same proportions, the $p$ value of the mixture distribution $p_m$ is the average $p$ value of the mixture components $\bar{p}_i$. In this setting, empirically choosing a single cutoff $x_m$ corresponding to a significance cutoff $\frac{\alpha}{n}$ based on the mixture distribution controls the FWER at the same level as setting gene-specific cutoffs $x_i$ such that all $\alpha_i = \alpha$ (the commonly applied approach).

The bound for family-wise error rate across all genes 1 to $n$ is given by Boole's inequality as used in the Bonferroni correction:

$$\text{FWER} = \Pr\left(\bigcup_{i=1}^{n}\left\{p_i \le \frac{\alpha}{n}\right\}\right) \le \sum_{i=1}^{n}\left\{\Pr\left(p_i \le \frac{\alpha}{n}\right)\right\} = \sum_{i=1}^{n}\frac{\alpha}{n} = \alpha \qquad (10)$$

Notably, tests can also be performed at different significance levels $\frac{\alpha_i}{n}$. If the average gene-specific alpha ($\bar{\alpha}_i$) is exactly $\alpha$, the bound for the FWER remains unchanged:

$$\sum_{i=1}^{n}\left\{\Pr\left(p_i \le \frac{\alpha_i}{n}\right)\right\} \overset{!}{=} \alpha \qquad (11)$$

This property has been used in the context of weighted Bonferroni correction, and related proofs apply[86]. Based on the formulae above, any cutoff $\alpha_m$ based on the mixture distribution corresponds to the average theoretical cutoff of the mixture components $\bar{\alpha}_i$. It follows that

$$\sum_{i=1}^{n}\left\{\Pr\left(p_m \le \frac{\alpha_m}{n}\right)\right\} = \sum_{i=1}^{n}\frac{\alpha_m}{n} = \sum_{i=1}^{n}\left\{\frac{\sum_{i=1}^{n}\frac{\alpha_i}{n}}{n}\right\} = \sum_{i=1}^{n}\frac{\alpha_i}{n} = \bar{\alpha}_i = \alpha_m \qquad (12)$$

The FWER is controlled at the same level as if we had performed tests using the gene-specific null distributions and set all $\alpha_i = \alpha$, or in

fact, any other values $\frac{\alpha_i}{n}$ that sum up to $\alpha^{86}$. Based on our experiments comparing $p$ values derived from the gene-specific null distributions to those derived from the mixture distribution, we show that $p_i \sim p_m$, and therefore $\alpha_i \sim \alpha_m$.

Additionally, we prevent false positives due to differences in $p_i$ and $p_m$ by performing gene-specific conditional tests for the genes which reach genome-wide significance using the mixture distribution. This means our approach may lose power: We can miss genes that would be significant using gene-specific null distributions, but not significant based on the mixture distribution.

## Software
Here we list software not otherwise mentioned in the manuscript. For the full list including version numbers consider the Reporting Summary. Our functional annotation and association testing pipeline (faatpipe) uses bcftools[87], bedtools[88], Plink[89], samtools[87], vcftools[90], htslib[91], biopython[92], pybedtools[93], pyranges[94], pysam (https://github.com/pysam-developers/pysam), and pysnptools (https://github.com/fastlmm/PySnpTools) to handle genomic ranges and genotype data.

Figures were produced using ggplot2[95], gplots[96], matplotlib[97], seaborn[98], and matplotlib_venn (https://github.com/konstantint/matplotlib-venn).

## Reporting summary
Further information on research design is available in the Nature Research Reporting Summary linked to this article.

## Data availability
Variant effect predictions for all variants in the 200k exome sequencing release are made available on github (https://github.com/HealthML/ukb-200k-wes-vep, v0.0.0, https://doi.org/10.5281/zenodo.6912352).

The weights for the DeepRiPe model used to predict the effects of variants on RBP-binding are available at https://github.com/HealthML/faatpipe/tree/master/data/deepripe_models. SpliceAI variant effect predictions are publicly made available by Illumina at https://basespace.illumina.com/s/otSPW8hnhaZR.

The Online Mendelian Inheritance in Man (OMIM®) database is publicly accessible through https://omim.org/. The ClinVar database is publicly accessible through https://www.ncbi.nlm.nih.gov/clinvar/. The NHGRI-EBI Catalog of human genome-wide association studies (GWAS catalog) is publicly accessible through https://www.ebi.ac.uk/gwas/. Phenoscanner, a database of human genotype-phenotype associations, is publicly available through http://www.phenoscanner.medschl.cam.ac.uk/. The 1000 Genomes phase 3 genotypes are publicly available at https://ftp.1000genomes.ebi.ac.uk/vol1/ftp/technical/working/20140708_previous_phase3/v5_vcfs/.

The genetic, phenotype, and covariate data are protected and are only available to researchers that have valid and approved research applications for these data within the UK Biobank (www.ukbiobank.ac.uk/).

## Code availability
A snakemake pipeline that allows reproducing results from this study is available on github (https://github.com/HealthML/faatpipe, v0.1.0, https://doi.org/10.5281/zenodo.6912198).

The implementation of statistical tests (score test, RLRT), including the python port of RLRsim[80] is available on github (https://github.com/HealthML/seak, v0.4.3, https://doi.org/10.5281/zenodo.6912202).

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

## Acknowledgements

The authors wish to thank Wolfgang Kopp for his valuable comments on the manuscript. This research has been conducted using the UK Biobank resource under application number 40502. This research has received funding from the German Federal Ministry of Education and Research (BMBF) in the projects SyReal (project number 01|S21069A) and KI-LAB-ITSE (project number 01|S19066), and the European Commission in the Horizon 2020 project INTERVENE (Grant agreement ID: 101016775).

## Author contributions

R.M., S.K., and C.L. conceived and designed the study. R.M., P.R., A.R.J., and S.K. performed initial prototyping. R.M. and P.R. wrote software to perform the statistical tests with guidance from U.O., S.K., and C.L. R.M. wrote the analysis pipeline with guidance from A.R.J., S.K., U.O., and C.L. R.M. carried out the statistical analyses with guidance from S.K., U.O., and C.L. M.G. supplied the weights for the DeepRiPe model and produced attribution maps. P.R. and R.M. queried GWAS databases and compared results with other studies. R.M., M.K., and C.L. contributed to the theoretical arguments regarding FWER control. R.M., P.R, S.K., and C.L. wrote the manuscript. All authors revised the manuscript.

## Funding

## Competing interests

The authors declare no competing interests.
