## [Peer Review File · Nature Communications]

Identifying interpretable gene-biomarker associations with functionally informed kernel-based tests in 190,000 exomesEditorial Note: This manuscript has been previously reviewed at another journal that is not operating a transparent peer review scheme. This document only contains reviewer comments and rebuttal letters for versions considered at *Nature Communications*. Mentions of the other journal have been redacted.

REVIEWER COMMENTS

Reviewer #1 (Remarks to the Author):

This is an updated version of a manuscript previously submitted to [redacted]. The authors conduct gene-level analyses of 30 biomarkers from the UK BioBank with rare variants. The main contributions are consideration of different types of variants, ancestry, and a novel modified kernel-based test. As before, I defer on the biological importance of the specific associations, but I continue to believe that this has the potential to be a nice contribution to the literature and that the comparisons of different types of tests for different types of variant categories to be of particular interest. Compared to the previous version, the authors have made efforts to clarify several raised points. However, I continue to have some questions regarding specific details of the work that I would like to see addressed. Comments follow in no order.

1. The text justifying the use of the pooled distribution is concerning (lines 711-714, supp methods). Specifically, the authors argue that the “average type-1 error rate” is controlled. However, the entire concept of FWER control is that it guarantees against any false rejections. Thus, the idea of average type I error rate seems inherently flawed – that is something that goes more along the lines of FDR control. That being said, the empirical results do seem quite impressive, and perhaps that is sufficient justification.
2. For the combined score-LRT approach, the authors screen first using the score test followed by the LRT, which is a nice approach. The threshold of 0.1 seems somewhat cautious, given that the actual alpha levels of interest are so much lower. Have the authors investigated whether a more aggressive threshold (e.g., 0.01 or lower) would be feasible? That would again speed up the approach by an order of magnitude.
3. If the score p-value is lower, do the investigators report that or the LRT p-value?
4. The qqplots still look slightly inflated in some cases (particularly at the 10^{-3} level or so), though better than I recall in the previous version of the manuscript. It would be helpful to know what changed.
5. I don't fully understand the rationale behind omitting variants based on ancestry. My understanding is that having diverse ancestral backgrounds could boost power in many cases, especially since what is rare in one population could be more common in another. The authors find this in their analyses (e.g., line 193). It would be helpful to understand this step some more.
6. When people argue about the score vs. LRT, I believe that this is under the mean model where variants affect the mean of the trait and where there are relatively sparse effects. However, based on the formulation here, the model is a variance model. I'm not sure that the same arguments hold.

Reviewer #2 (Remarks to the Author):

The authors addressed my previous comments. The results and methods are not exciting but serve as a useful resource for the field. I have no further comments.

Reviewer #3 (Remarks to the Author):

The authors have addressed my concerns.

REVIEWER COMMENTS

Reviewer #1 (Remarks to the Author):

This is an updated version of a manuscript previously submitted to [redacted]. The authors conduct gene-level analyses of 30 biomarkers from the UK BioBank with rare variants. The main contributions are consideration of different types of variants, ancestry, and a novel modified kernel-based test. As before, I defer on the biological importance of the specific associations, but I continue to believe that this has the potential to be a nice contribution to the literature and that the comparisons of different types of tests for different types of variant categories to be of particular interest. Compared to the previous version, the authors have made efforts to clarify several raised points. However, I continue to have some questions regarding specific details of the work that I would like to see addressed. Comments follow in no order.

We thank the reviewer for the constructive feedback and attention to detail.

1. The text justifying the use of the pooled distribution is concerning (lines 711-714, supp methods). Specifically, the authors argue that the “average type-1 error rate” is controlled. However, the entire concept of FWER control is that it guarantees against any false rejections. Thus, the idea of average type I error rate seems inherently flawed – that is something that goes more along the lines of FDR control. That being said, the empirical results do seem quite impressive, and perhaps that is sufficient justification.

In the first revision, the reviewer asked if it is valid to use a pooled null distribution for the LRT as opposed to a gene-specific null distribution (point 2 in last revision) and asked whether this could be related to test statistic inflation for kernel-based tests (point 3 in last revision).

We clarified that we use pooled distributions that are specific to each variant-type to test-type and biomarker combination (point 2 in last revision), and empirically showed that this approximation yields consistent results (point 3 in last revision). We are glad that the reviewer finds these results convincing.

With regards to the FWER, we used the term “average type-1 error rate” because the p-value of the mixture distribution is simply the average p-value of the mixture components if the components are sampled to equal proportions. Using Bonferroni correction to bound the FWER at a specific level α_m based on the mixture implicitly asserts that the corresponding theoretical thresholds for the mixture components $\frac{\alpha_i}{n}$ (in our case, the single genes) sum up to α_m and therefore the FWER is guaranteed. This property has been used in the context of weighted Bonferroni correction, and related proofs apply¹. Our empirical results mentioned above further show that under our modelling assumptions the gene-specific p-values are close to the mixture p-values.

Additional to any theoretical guarantees, we prevent false positives due to differences of the pooled and gene-specific null distributions by performing gene-specific conditional tests for the genes which reach genome-wide significance. Because of this, our approach can lose power: We can miss genes that would be significant using gene-specific null distributions, but not significant based on the mixture distribution.

We have removed the sentence in question from the supplementary methods and added a paragraph “FWER control for the pooled null distribution” in the supplementary methods to clarify the points regarding the FWER.

2. For the combined score-LRT approach, the authors screen first using the score test followed by the LRT, which is a nice approach. The threshold of 0.1 seems somewhat cautious, given that the actual alpha levels of interest are so much lower. Have the authors investigated whether a more aggressive threshold (e.g., 0.01 or lower) would be feasible? That would again speed up the approach by an order of magnitude.

Yes, we have investigated the use of smaller thresholds. These findings are summarized in Figures S6 and S7. Figure S7 shows that for splice and missense variants the sLRT identifies more associations than the score test alone, and, importantly, that the associations found by the score test are, with 1-2 exceptions, a subset of those identified by the sLRT. Figure S6 shows the cutoff for the score-test p-values (x-axis) against the number of associations identified by the sLRT. Towards the very left of the x-axis, any hit identified by the combination of score- and likelihood ratio test would already have been significant using score tests alone. As we lower the threshold from left to right, the number of hits identified by the sLRT increases. These figures show that more stringent thresholds such as 0.01 and 0.001 are indeed possible, at the expense of only a few missed significant associations per variant category.

3. If the score p-value is lower, do the investigators report that or the LRT p-value?

If the LRT is performed, the p-value for the test always comes from the LRT. We mention this in the section “Combined likelihood ratio and score tests (sLRT)”.

4. The qqplots still look slightly inflated in some cases (particularly at the 10^{-3} level or so), though better than I recall in the previous version of the manuscript. It would be helpful to know what changed.

We address this comment together with the one below, as they are related.

5. I don't fully understand the rationale behind omitting variants based on ancestry. My understanding is that having diverse ancestral backgrounds could boost power in many cases, especially since what is rare in one population could be more common in another. The authors find this in their analyses (e.g., line 193). It would be helpful to understand this step some more.

During revisions, we performed the analysis for a strict European ancestry subset and found that test statistics were less inflated for some phenotypes. We concluded that the inflation we observed might be due to residual confounding not captured by genetic principal components (we had already excluded related individuals). Ancestry-based variant pruning was primarily used to mitigate this confounding, as was the inclusion of ancestry variables from the ancestry prediction model. Because the sample size for other ancestry groups in the UK Biobank are relatively small, the filter we implemented mainly removed variants that were significantly more common in the other ancestries compared to the European ancestry subset.

We agree, such a filter could potentially remove causal variants and therefore result in a loss of power. However, the fact that these variants are much more common in other ancestry groups can be seen as evidence against strong detrimental causal effects. This assumption is commonly made in rare-variant association studies (e.g., by up-weighting rarer variants). Other studies have used similar approaches, e.g., removing variants above an allele frequency threshold in any reference population². The highlighted example in *PIEZO1* might represent somewhat of an outlier, i.e., a variant that is common in one ancestry group, yet has a strong causal detrimental effect. It is unclear why this variant was able to reach such a high allele frequency in that population, especially as it has been linked to hereditary dehydrated stomatocytosis³ (in a family study we have only found after resubmission, which we now mention and added to the references).

Other methods account for confounding by population structure by including a random effect capturing relatedness in the null model⁴ or related approaches⁵. Yet, the specialized available software does not support the variant-category specific kernels we propose, or the use of the LRT. For the likelihood-ratio test, the fast sampling of test statistics that our analysis relies on cannot account for an additional random effect. We see expanding these methods as directions for future research that are outside of the scope of our analysis.

We used ancestry-based variant pruning as a simple alternative to prevent test statistic inflation that worked for us in practice and didn't require us to completely abandon or re-write our existing analysis pipeline and software. Considering the strong external evidence supporting the significant associations, we concluded this approach was sufficiently stringent, especially for the comparisons made in the manuscript. We added a paragraph in the conclusion to clarify these points (line 315).

Any remaining inflation in the tails could either be due to weak signals in some genes, weak residual confounding, or both. In the absence of a ground truth, we cannot conclude either.

6. When people argue about the score vs. LRT, I believe that this is under the mean model where variants affect the mean of the trait and where there are relatively sparse effects. However, based on the formulation here, the model is a variance model. I'm not sure that the same arguments hold.

The references we refer to regarding score vs. LRT examined the variance model^{6,7}. In our study, we treat the model in which the variants affect the mean of the trait as a special case of the variance model, as was previously proposed⁸. The mean model assumes that all variants tested affect the trait in the same direction. We explain the corresponding formulation under the variance model in section “Statistical models and tests”, line 418.

Reviewer #2 (Remarks to the Author):

The authors addressed my previous comments. The results and methods are not exciting but serve as a useful resource for the field. I have no further comments.

We thank the reviewer for their suggestions to improve our manuscript.

Reviewer #3 (Remarks to the Author):

The authors have addressed my concerns.

We thank the reviewer for their suggestions to improve our manuscript.

References:

1. Genovese, C. R., Roeder, K. & Wasserman, L. False discovery control with p-value weighting. *Biometrika* **93**, 509–524 (2006).
2. Cirulli, E. T. *et al.* Genome-wide rare variant analysis for thousands of phenotypes in over 70,000 exomes from two cohorts. *Nat. Commun.* **11**, (2020).
3. Picard, V. *et al.* Clinical and biological features in PIEZO1-hereditary xerocytosis and gardos channelopathy: A retrospective series of 126 patients. *Haematologica* **104**, 1554–1564 (2019).
4. Zhou, W. *et al.* Scalable generalized linear mixed model for region-based association tests in large biobanks and cohorts. *Nat. Genet.* **52**, 634–639 (2020).
5. Mbatchou, J. *et al.* Computationally efficient whole-genome regression for quantitative and binary traits. *Nat. Genet.* **53**, 1097–1103 (2021).
6. Listgarten, J. *et al.* A powerful and efficient set test for genetic markers that handles confounders. *Bioinformatics* **29**, 1526–1533 (2013).
7. Lippert, C. *et al.* Greater power and computational efficiency for kernel-based association testing of sets of genetic variants. *Bioinformatics* **30**, 3206–3214 (2014).
8. Lee, S., Wu, M. C. & Lin, X. Optimal tests for rare variant effects in sequencing association studies. *Biostatistics* **13**, 762–775 (2012).

Reviewer #1 (Remarks to the Author):

I have no outstanding concerns. The authors have done an excellent job clarifying.

REVIEWERS' COMMENTS

Reviewer #1 (Remarks to the Author):

I have no outstanding concerns. The authors have done an excellent job clarifying.

We thank the reviewer for their suggestions.